# Towards Metamerism via Foveated Style Transfer

**Arturo Deza**[1,4], **Aditya Jonnalagadda**[3], **Miguel P. Eckstein**[1,2,4]
[1] Dynamical Neuroscience, [2]Psychological and Brain Sciences,
[3]Electric and Computer Engineering, [4] Institute for Collaborative Biotechnologies
UC Santa Barbara, CA, USA
`deza@dyns.ucsb.edu,aditya_jonnalagada@ece.ucsb.edu,eckstein@psych.ucsb.edu`

## Abstract

The problem of *visual metamerism* is defined as finding a family of perceptually indistinguishable, yet physically different images. In this paper, we propose our NeuroFovea metamer model, a foveated generative model that is based on a mixture of peripheral representations and style transfer forward-pass algorithms. Our gradient-descent free model is parametrized by a foveated VGG19 encoder-decoder which allows us to encode images in high dimensional space and interpolate between the content and texture information with adaptive instance normalization anywhere in the visual field. Our contributions include: 1) A framework for computing metamers that resembles a noisy communication system via a foveated feed-forward encoder-decoder network – We observe that metamerism arises as a byproduct of noisy perturbations that partially lie in the perceptual null space; 2) A perceptual optimization scheme as a solution to the hyperparametric nature of our metamer model that requires tuning of the image-texture tradeoff coefficients everywhere in the visual field which are a consequence of internal noise; 3) An ABX psychophysical evaluation of our metamers where we also find that the rate of growth of the receptive fields in our model match V1 for reference metamers and V2 between synthesized samples. Our model also renders metamers at roughly a second, presenting a ×1000 speed-up compared to the previous work, which allows for tractable data-driven metamer experiments.

## 1 Introduction

The history of metamers originally started through color matching theory, where two light sources were used to match a test light's wavelength, until both light sources are indistinguishable from each other producing what is called a *color metamer*. This leads to the definition of visual metamerism: when two physically different stimuli produce the same perceptual response (See Figure 1 for an example). Motivated by Balas et al. (2009)'s work of local texture matching in the periphery as a mechanism that explains visual crowding, Freeman & Simoncelli (2011) were the first to create such point-of-fixation driven metamers through such local texture matching models that tile the entire visual field given log-polar pooling regions that simulate the V1 and V2 receptive field sizes, as well as having global image statistics that match the metamer with the original image. The essence of their algorithm is to use gradient descent to match the local texture (Portilla & Simoncelli (2000)) and image statistics of the original image throughout the visual field given a point of fixation until convergence thus producing two images that are perceptually indistinguishable to each other.

However, metamerism research currently faces 2 main limitations: The first is that metamer rendering faces no unique solution. Consider the potentially trivial examples of having an image *I* and its metamer *M* where all pixel values are identical except for one which is set to zero (making this difference unnoticeable), or the case where the metameric response arises from an imperceptible equal perturbation across all pixels as suggested in Johnson et al. (2016); Freeman & Simoncelli (2011). This is a concept similar to Just Noticeable Differences (Lubin (1997); Daly (1992)). However, like the work of Freeman & Simoncelli (2011); Keshvari & Rosenholtz (2016); Rosenholtz et al. (2012); Balas et al. (2009), we are interested in creating point-of-fixation driven metamers, which create images that preserve information in the fovea, yet lose spatial information in the periphery such that this loss is unnoticeable contingent of a point of fixation (Figure 1). The second issue is that the current state of the art for a full field of view rendering of a 512px×512px metamer takes 6 hours for a grayscale image and roughly a day for a color image. This computational constraint makes data-

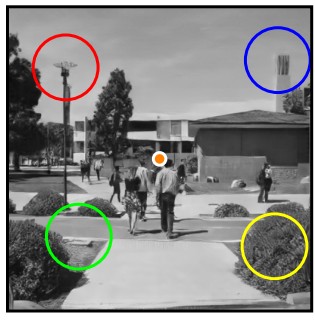 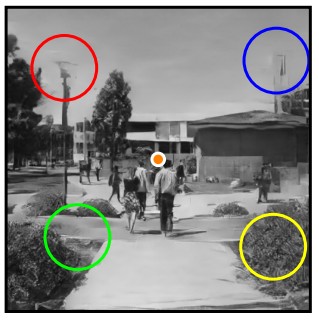 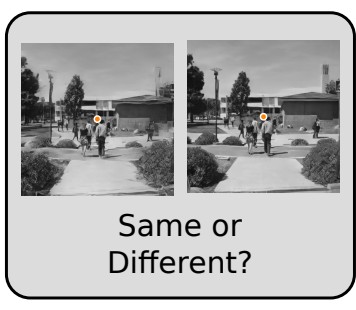

Image                    Metamer

Figure 1: Two visual metamers are physically different images that when fixated on the orange dot (center), should remain perceptually indistinguishable to each other for an observer. Colored circles highlight different distortions in the visual field that observers do not perceive in our model.

driven experiments intractable if they require thousands of metamers. From a practical perspective, creating metamers that are quick to compute may lead to computational efficiency in rendering of VR foveated displays and creation of novel neuroscience experiments that require metameric stimuli such as gaze-contingent displays, or metameric videos for fMRI, EEG, or Eye-Tracking.

We think there is a way to capitalize metamer understanding and rendering given the developments made in the field of *style transfer*. We know that the original model of Freeman & Simoncelli (2011) consists of a local texture matching procedure for multiple pooling regions in the visual field as well as global image content matching. If we can find a way to perform localized style transfer with proper texture statistics for all the pooling regions in the visual field, and if the metamerism via texture-matching hypothesis is correct – we can in theory successfully render a metamer.

Within the context of style transfer, we would want a complete and flexible framework where a *single* network can encode *any* style (or texture) without the need to re-train, and with the power of producing style transfer with a single forward pass, thus enabling real-time applications. Furthermore, we would want such framework to also control for spatial and scale factors (Gatys et al. (2017)) to enable foveated pooling (Akbas & Eckstein (2017); Deza & Eckstein (2016)) which is critical in metamer rendering. The very recent work of Huang & Belongie (2017), provides such framework through adaptive instance normalization (AdaIN), where the content image is stylized by adjusting the mean and standard deviation of the channel activations of the encoded representation to match with the style. They achieve results that rival those of Ulyanov et al. (2016); Johnson et al. (2016), with the added benefit of not being limited to a single texture in a feed-forward pipeline.

In our model: we stack a peripheral architecture on top of a VGGNet (Simonyan & Zisserman (2015)) in its encoded feature space, to map an image into a perceptual space. We then add internal noise in the encoded space of our model as a characterization that perceptual systems are noisy. We find that inverting such modified image representation via a decoder results in a metamer. This breaks down our model into a foveated feed-forward 'auto' style transfer network, where the input image plays the role both of the content and the style, and internal network noise (stylized with the content statistics) serves as a proxy for intrinsic image texture. While our model uses AdaIN for style transfer and a VGGNet for texture statistics, our pipeline is extendible to other models that successfully execute style transfer and capture proper texture statistics (Ustyuzhaninov et al. (2017)).

## 2 DESIGN OF THE NEUROFOVEA MODEL

To construct our metamer we propose the following statement: A metamer $M$ can be rendered by transferring $k$ *localized* styles over a content image $I$, controlled by a set of style-to-content ratios $\alpha_i$ for every pooling region ($i$-th receptive field). More formally, our goal is to find a Metamer function $\mathbf{M}(\circ) : I \to M$, where an input image $I \in \mathbb{R}^L$ is fed through a VGG-Net encoder $\mathcal{E}(\cdot) : \mathbb{R}^L \to \mathbb{R}^D$ which is both the content and the style image, to produce the content feature $\mathbf{C} \in \mathbb{R}^D$, where $\mathbf{C} = \mathcal{E}(I)$ as shown in Figure 2. Let $L = C \times H \times W$, and $D = C' \times H' \times W'$ where $\{C, C'\}, \{H, H'\}, \{W, W'\}$ are the image/layer channels, height, width given the convolutional structure of the encoder (we drop fully connected layers). A noise patch colored via ZCA (Bell & Sejnowski (1995)) to match the

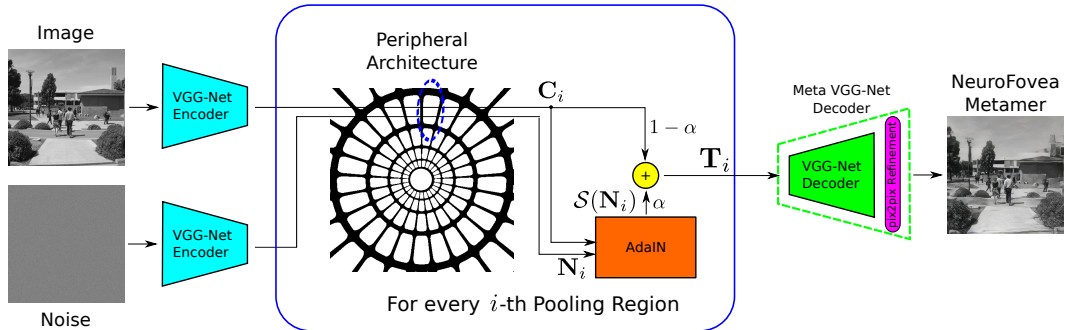

Figure 2: The NeuroFovea metamer generation schematic: An input image and a noise patch are fed through a VGG-Net encoder into a new feature space. Through spatial control we can produce an interpolation for each pooling region in such feature space between the stylized-noise (texture), and the content (the input image). This is how we successfully impose both global image and local texture-like constraints in every pooling region. The metamer is the output of the pooled (and interpolated) feature vector through the Meta VGG-Net Decoder.

content image's mean and variance $\mathcal{N} \sim (\mu_I, \sigma_I^2) \in \mathbb{R}^L$ is also fed through the same VGG-Net encoder producing the noise feature $\mathbf{N} \in \mathbb{R}^D$, where $\mathbf{N} = \mathcal{E}(\mathcal{N})$. This is the internal perceptual noise of the system which will later on serve us as a proxy for texture encoding. These vectors are masked through spatial control *a la* Gatys et al. (2017), and the noise is stylized via $\mathcal{S}(\cdot) : \mathbb{R}^D \to \mathbb{R}^D$ with the content which encodes the texture representation of the content in the feature space through Adaptive Instance Normalization (AdaIN). A target feature $\mathbf{T}_i \in \mathbb{R}^D$ is defined as an interpolation between the stylized noise $\mathcal{S}(\mathbf{N}_i)$ and the content $\mathbf{C}_i$ modulated by $\alpha$, in the feature space $\mathbb{R}^D$ for every $i$-th pooling region:

$$\mathbf{T}_i(I|\mathcal{N};\alpha) = (1-\alpha)\mathbf{C}_i(I) + \alpha\mathcal{S}(\mathbf{N}_i) \tag{1}$$

In other words, in our quest to probe for metamerism, we are finding an intermediate representation (the convex combination) between two vectors representing the image and its texturized version (the stylized noise) in $\mathbb{R}^D$ per pooling region as seen in Figure 3. Within the framework of style transfer, we could think of this as a content-vs-style or structure-vs-texture tradeoff, since the style and the content image are the same. Similar interpolations have been explored in Hénaff & Simoncelli (2016) via a joint pixel and network space minimization. The final target feature vector $\mathbf{T}$ is the masked sum of every $\mathbf{T}_i$ with spatial control masks $w_i$ s.t. $\mathbf{T} = \sum w_i \mathbf{T}_i$. The metamer is the output of the Meta VGG-Net decoder $\mathcal{D}(\cdot)$ on $\mathbf{T}$, where the decoder receives only *one* vector ($\mathbf{T}$) and produces a global decoded output. Our Meta VGG-Net Decoder compensates for small artifacts by stacking a *pix2pix* Isola et al. (2017) U-Net refinement module which was trained on the Encoder-Decoder outputs to map to the original high resolution image. Figure 2 fully describes our model, and the metamer transform is computed via:

$$\mathbf{M}(I|\mathcal{N};\bar{\alpha}) = \mathcal{D}(\mathcal{E}_\Sigma(I|\mathcal{N};\bar{\alpha})) = \mathcal{D}(\sum_{i=1}^{k} w_i[(1-\alpha_i)\mathcal{E}_i(I) + \alpha_i\mathcal{S}(\mathcal{E}_i(\mathcal{N}))]) \tag{2}$$

where $\mathcal{E}_\Sigma$ is the foveated encoder that is defined as the sum of encoder outputs over all the $k$ pooling regions (our spatial controls masks $w_i$) in the visual field. Note that the decoder was not trained to generate metamers, but rather to invert the encoded image and act as $\mathcal{E}^{-1}$. It happens to be the

| $\alpha = 0.0$ | $\alpha = 0.2$ | $\alpha = 0.4$ | $\alpha = 0.6$ | $\alpha = 0.8$ | $\alpha = 1.0$ |
| --- | --- | --- | --- | --- | --- |

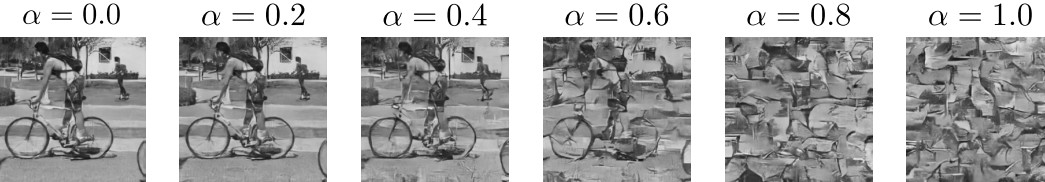

Figure 3: Interpolating between an image's intrinsic content and texture via a convex combination in the output of the VGG19 Encoder $\mathcal{E}$. Here we are treating the patch as a single pooling region. In our model, this interpolation given Eq. 1 is done for every pooling region in the visual field.

case that perturbing the encoded representation in the direction of the stylized noise by an amount specified by the size of the pooling regions, outputs a metamer. Additional specifications and training of our model can be seen in the Supplementary Material.

## 2.1 MODEL INTERPRETABILITY

Within the framework of metamerism where distortions lie on the perceptual null space as proposed initially in color matching theory, and also in Freeman & Simoncelli (2011) for images, we can think of our model as a direct transform that is maximizing how much information to discard depending on the texture-like properties of the image and the size of the receptive fields. Consider the following: if our interpolation is projected from the encoded space to the perceptual space via $P$, from Eq. 1 we get $P\mathbf{T}_i = P(1-\alpha)\mathbf{C}_i(I) + P(\alpha)\mathcal{S}(\mathbf{N}_i)$, it follows that for each receptive field:

$$P\underbrace{\mathbf{T}_i}_{\text{metamer}} = P\underbrace{\mathbf{C}_i}_{\text{image}} + P\underbrace{\alpha(\mathcal{S}^\perp(\mathbf{N}_i) + \mathcal{S}^\|(\mathbf{N}_i))}_{\text{distortion}} \tag{3}$$

by decomposing $\mathcal{S}(\mathbf{N}_i) - \mathbf{C}_i = \mathcal{S}^\perp(\mathbf{N}_i) + \mathcal{S}^\|(\mathbf{N}_i)$, where $\mathcal{S}^\|$ is the projection of the difference vector on the perceptual space, and $\mathcal{S}^\perp(\mathbf{N}_i)$ is the orthogonal component perpendicular to such vector which lies in the perceptual null space ($P\mathcal{S}^\perp(\mathbf{N}_i) = \vec{0}$). The value of these components will change depending on the location of $\mathbf{C}_i$ and $\mathcal{S}(\mathbf{N}_i)$, and the geometry of the encoded space. If $\|\mathcal{S}^\|(\mathbf{N}_i)\|_2^2 < \epsilon$, (i.e. the image patch has strong texture-like properties), then $\alpha$ can vary above its critical value given that $\mathcal{S}^\perp(\mathbf{N}_i)$ is in the null space of $P$ and the distortion term will still be small; but if $\|\mathcal{S}^\|(\mathbf{N}_i)\|_2^2 > \epsilon$, $\alpha$ can not exceed its critical value for the metamerism condition to hold ($P\mathbf{T}_i \approx P\mathbf{C}_i$). Thus our interest is in computing the maximal *average* amount of distortion (driven by $\alpha$) given human sensitivity before observers can tell the difference. This is illustrated in Figure 4 via the blue circle around $\mathbf{C}_i$ in the perceptual space which shows the *metameric boundary* for any distortion.

One can also see the resemblance of the model to a noisy communication system in the context of information theory. The information source is the image $I$, the transmitter and the receiver are the encoder and decoders ($\mathcal{E}, \mathcal{D}$) respectively, and the noise source is the encoded noise patch $\mathcal{E}(\mathcal{N})$ imposing texture distortions in the visual field, and the destination is the metamer $M$. Highlighting this equivalence is important as metamerism can also be explored within the context of image compression and rate-distortion theory as in Ballé et al. (2017). Such approaches are beyond the scope of this paper, however they are worth exploring in future work as most metamer models purely involve texture and image analysis-synthesis matching paradigms that are gradient-descent based.

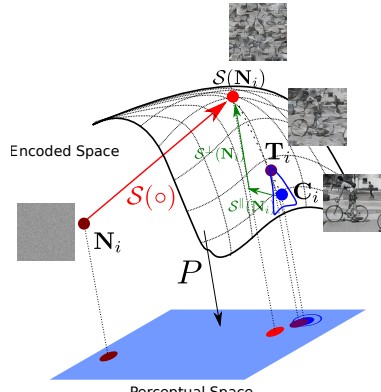

Figure 4: Perceptual Projection.

## 3 HYPERPARAMETERIC NATURE OF OUR MODEL

Similar to our model, the Freeman & Simoncelli (2011) model (hereto be abbreviated FS) requires a scale parameter $s$ which controls the rate of growth of the receptive fields as a function of eccentricity. This parameter should be maximized such that an upperbound for perceptual discrimination is found. Given that texture and image matching occurs in each one of the pooling regions: a high scaling factor will likely make the image rapidly distinguishable from the original as distortions are more apparent in the periphery. Conversely, a low scaling factor might gaurantee metamerism even if the texture statistics are not fully correct given that smaller pooling regions will simulate weak effects of crowding. Low scaling factors in that sense are potentially uninteresting – it is the value up until humans can tell the difference that is critical (Lubin (1997)). FS set out to find such critical value via a psychophysical experiment where they perform the following single-variable optimization to find such upper bound:

$$s_0 = \arg\max_s \ \mathbb{E}[d'(s|\theta_{obs})] \tag{4}$$

s.t. $0 < d'(s|\theta_{obs}) < \epsilon$, where $d' = \Phi^{-1}(\text{HR}) - \Phi^{-1}(\text{FA})$ is the index of detectability for each observer $\theta_{obs}$, $\Phi$ is the cumulative of the gaussian distribution, and HR and FA are the hit rate and false alarm rates as defined in Green & Swets (1966). However, our model is different in regards to a set of

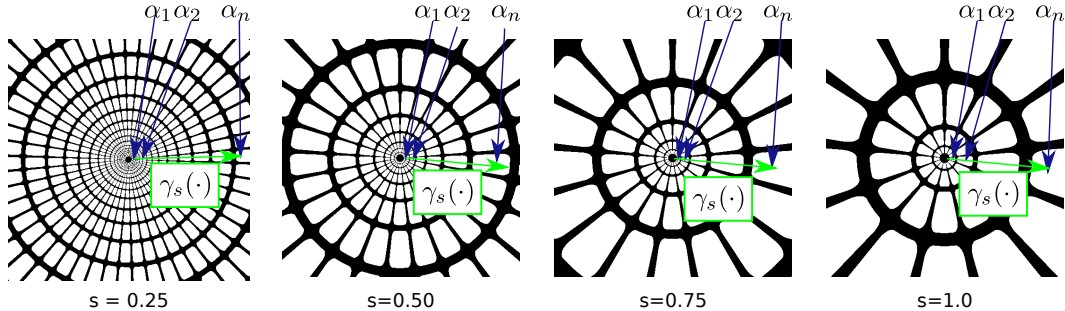

Figure 5: Potential issues of psychophysical intractability for the joint estimation of ($s$) and $\gamma(\cdot)$ as described by our model. Running a psychophysical experiment that runs an exhaustive search for upper bounds for the scale and distortion parameters for every receptive field is intractable. The goal of Experiment 1 is to solve this intractabitilty posed formally in Eq. 6 via a simulated experiment.

hyperparameters $\bar{\alpha}$ that we must estimate everywhere in the visual field as summarized by the $\gamma$ function, where we assume $\alpha$ to be tangentially isotropic:

$$\alpha = \gamma(\circ; s) \tag{5}$$

where each $\alpha$ represents the maximum amount of distortion (Eq. 1) that is allowed for every receptive field in the visual periphery before an observer will notice. At a first glance, it is not trivial to know if $\alpha$ should be a function of scale, retinal eccentricity, receptive field size, image content or potentially a combination of the before-mentioned (hence the $\circ$ in the $\gamma$ function's argument).

Thus, the motivation of $\alpha$ seems uncertain and perhaps un-necessary from the Occam's razor perspective of model simplicity. This raises the question: Why does the FS model not require any additional hyperparameters, requiring only a single scale ($s$) parameter? The answer lies in the nature of their model which is gradient descent based and where local texture statistics are matched for every pooling region in the visual field, while preserving global image structural information. When such condition is reached, no further synthesis steps are required as it is an equilibrium point. Indeed, the experiments of Wallis et al. (2016) have shown that images do not remain metameric if the structural information of a pooling region is discarded while purely retaining the texture statistics of Portilla & Simoncelli (2000). This motivates the purpose of $\alpha$ where we interpolate between structural and texture representation. Thus our goal is to find that equilibirum point in one-shot, given that our model is purely feed-forward and requires no gradient-descent (Eq. 2). At the expense of this artifice, we run into the challenge of facing a multi-variable optimization problem that has the risk of being psychophysically intractable. Analogous to FS, we must solve:

$$s_0, \bar{\alpha_0} = \underset{s, \bar{\alpha}}{\arg\max} \; \mathbb{E}[d'(s, \bar{\alpha}|\theta_{obs})] \tag{6}$$

s.t. $0 < d'(s, \bar{\alpha}|\theta_{obs}) < \epsilon$. Figure 5 shows the potential intractability: each observer would have to run multiple rounds of an ABX experiment for a collection of many scales and $\alpha$ values for each location in the visual field. Consider: ($S$ scales) $\times$ ($k$ pooling regions) $\times$ ($\alpha_m$ step size for each $\alpha$) $\times$ ($N$ images) $\times$ ($w$ trials): $S k N \alpha_m w$ trials per observer.

We will show in Experiment 1 that one solution to Eq. 6 is to find a relationship between each set of $\alpha$'s and the scale, expressed via the $\gamma$ function. This requires a two stage process: 1) Showing that such $\gamma$ exists; 2) Estimate $\gamma$ given $s$. If this is achieved, we can relax the multi-variable optimization into a single variable optimization problem, where $0 < d'(s, \gamma(\circ; s)|\theta_{obs}) < \epsilon$, and:

$$s_0 = \underset{s}{\arg\max} \; \mathbb{E}[d'(s, \gamma(\circ; s)|\theta_{obs})] \tag{7}$$

## 4 EXPERIMENTS

The goal of Experiment 1 is to estimate $\gamma$ as a function of $s$ via a computational simulation as a proxy for running human psychophysics. Once it is computed, we have reduced our minimization to a tractable single variable optimization problem. We will then proceed to Experiment 2 where we will perform an ABX experiment on human observers by varying the scale to render visual metamers as originally proposed by FS. We will use the images shown in Figure 6 for both our experiments.

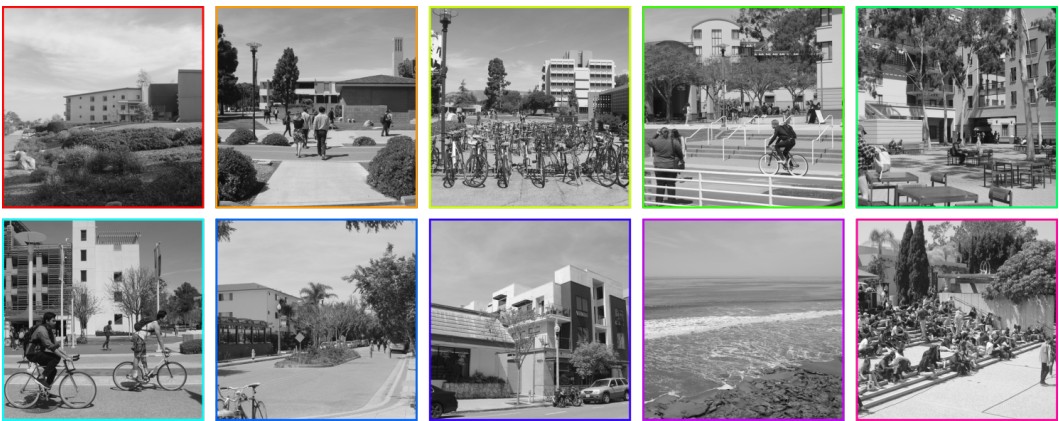

Figure 6: A color-coded collection of images used in our experiments.

### 4.1 EXPERIMENT 1: ESTIMATION OF MODEL HYPERPARAMETERS VIA PERCEPTUAL OPTIMIZATION

**Existence and shape of $\gamma$:** Given some biological priors, we would like $\gamma$ to satisfy these properties:

1. $\gamma : Z \to \alpha$ s.t. $Z \in [0, \infty), \alpha \subset [0, 1]$, where $z \in Z$ is parametrized by the size (radius) of each receptive field (pooling region) which grows with eccentricity in humans.

2. $\gamma$ is continuous and monotonically non-decreasing since more information should not be gained given larger crowding effects as receptive field size increases in the periphery.

3. $\gamma$ has a unique zero at $\gamma(0) = 0$. Under ideal assumptions there is no loss of information in the fovea, where the size of the receptive fields asymptotes to zero.

Indeed, we found that $\gamma$ is sigmoidal, and is a function of $z$, parametrized by $s$:

$$\gamma(z; s) = a + \frac{b}{c + \exp(-dz)} = -1 + \frac{2}{1 + \exp(-d(s)z)} \tag{8}$$

**Estimation of $\gamma$:** To numerically estimate the amount of $\alpha$-noise distortion for each receptive field in our metamer model we need to find a way to simulate the perceptual loss made by a human observer when trying to discriminate between metamers and original images. We will define a perceptual loss $\mathcal{L}$ that has the goal of matching the distortions via SSIM of a gradient descent based method such as the FS metamers, and the NeuroFovea metamers (NF) with their reference images – a strategy similar to Laparra et al. (2017) used for perceptual rendering. We chose SSIM as it is a standard IQA metric that is monotonic with human judgements, although other metrics such as MS-SSIM and IW-SSIM show similar tuning properties for $\gamma$ as shown in the Supplementary Material. Indeed the *reference* image $I'$ for the NF metamer is limited by the autoencoder-like nature of the model where the bottleneck usually limits perfect reconstruction s.t. $I' = \mathcal{D}(\mathcal{E}(I))|_{(\alpha=0)}$, where $I' \to I$, and they are only equal if the encoder-decoder pair

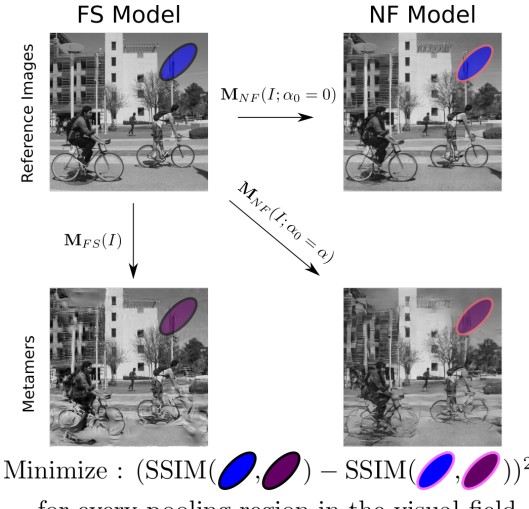

Minimize : $(\text{SSIM}(\text{⬤},\text{⬤}) - \text{SSIM}(\text{⬤},\text{⬤}))^2$
for every pooling region in the visual field

Figure 7: Perceptual optimization.

$(\mathcal{E}, \mathcal{D})$ allows for lossless compression. Since we can not define a direct loss function $\mathcal{L}$ between the metamers, we will need their reference images to define a convex surrogate loss function $\mathcal{L}_R$. The goal of this function should be to match the perceptual loss of both metamers *for each receptive field*

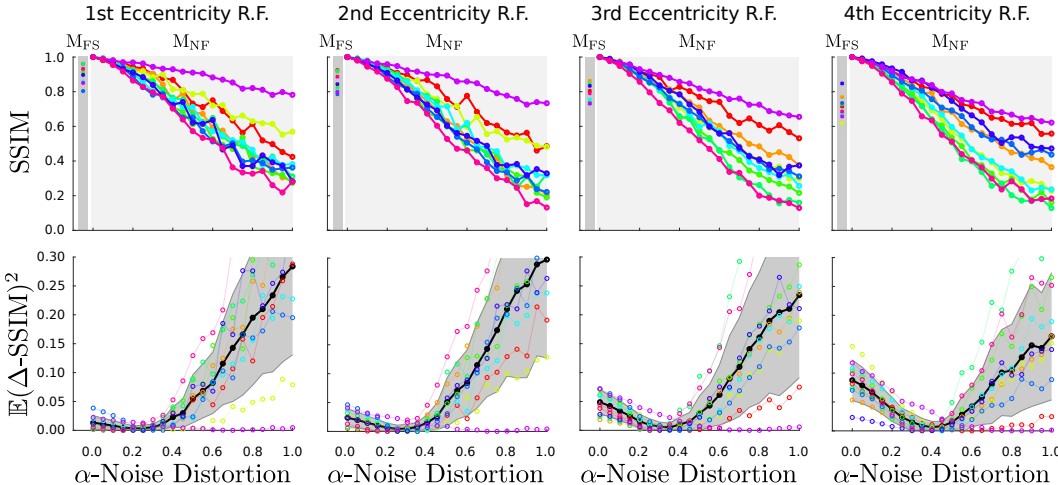

Figure 8: The result of each SSIM (top) for Experiment 1 for a scale of $s = 0.3$ where we find the critical $\alpha$ for each receptive field ring as we minimize $\mathbb{E}(\Delta\text{-SSIM})^2$ (bottom). $\mathbb{E}(\Delta\text{-SSIM})^2$ is minimized by matching the perceptual distortion of the Freeman & Simoncelli (2011) ($M_{FS}$) and NeuroFovea ($M_{NF}$) metamers in Eq. 9. Each color represents a different $512 \times 512$ image trajectory, the black line (bottom) shows the average. Only the first 4 eccentricity dependent receptive fields are shown.

$k$ when compared to their reference images: the original image $I$ for the FS model, and the decoded image $I'$ for the NF model:

$$\mathcal{L}_R(\alpha|k) = \mathbb{E}(\Delta\text{-SSIM})^2 = \frac{1}{N}\sum_{j=1}^{N}(\text{SSIM}(M_{FS}^{(j,k)}, I^{(j,k)}) - \text{SSIM}(M_{NF}^{(j,k)}(\gamma_s), I'^{(j,k)}))^2 \qquad (9)$$

and $\alpha_i$ should be minimized for each $k$ pooling region via: $\alpha_0 = \arg\min_\alpha \mathcal{L}_R(\alpha|k)$ for the collection of $N$ images. The intuition behind this procedure is shown in Figure 7. Note that if $I' = I$, *i.e.* there is perfect lossless compression and reconstruction given the choice of encoder and decoder, then the optimization is performed with reference to the same original image. This is an important observation as the reconstruction capacity of our decoder is limited despite $\mathbb{E}(\text{MS-SSIM}(I, I')) = 0.86 \pm 0.04$. Only using the original image in the optimization yields poor local minima at $\alpha = 0$. Despite such limitation, we show that reference metamers can still be achieved for our lossy compression model.

**Results:** A collection of 10 images were used in our experiments. We then computed the SSIM score for each FS and NF image paired with their reference image across each receptive field (R.F.) and averaged those that belonged to the same retinal eccentricity. Figure 8 (top) shows these results, as well as the convex nature of the loss function displayed in the bottom. This procedure was repeated for all the eccentricity-dependent receptive fields for a collection of 5 values of scale: $\{0.3, 0.4, 0.5, 0.6, 0.7\}$. A sigmoid to estimate $\gamma$ was then fitted to each $\alpha$ per R.F. parametrized by scale via least squares. This gave us a collection of $d$ values that control the slope rate of the sigmoid (Eq. 8). These were $d : \{1.240, 1.196, 1.363, 1.311, 1.355\}$ respectively per scale, and $\{d\} = 1.281$ for the ensemble of all scales. We then conducted a 10000 sample permutation test between the pair of $(z_s, \alpha_s)$ points per scale and the ensemble of points across all scales $(\{z\}, \{\alpha\})$ that verified that their variation is statistically non-significant ($p \geq 0.05$). Figure 9 illustrates the results from such procedure. We can conclude that the parameters of $\gamma$ do not vary as we vary scale. In other words, the $\alpha = \gamma(z)$ function is fixed, and the scale parameter itself which controls receptive field size will implicitly modulate the maximum $\alpha$-noise distortion with a unique $\gamma$ function. If the scale factor is small, the maximum noise distortion in the far periphery will be small and *vice versa* if the scale is large. We should point out that Figure 9 might suggest that the maximal noise distortion is contingent on image content as the scores are not uniform tangentially for the receptive fields that lie on the same eccentricity ring. Indeed, we did simplify our model by computing an average and fitting the sigmoid. However, computing an average should approximate the maximal distortion for the receptive field size on that eccentricity in the *perceptual space* for the human observer *i.e.* the metameric boundary. We elaborate more on this idea in the discussion section.

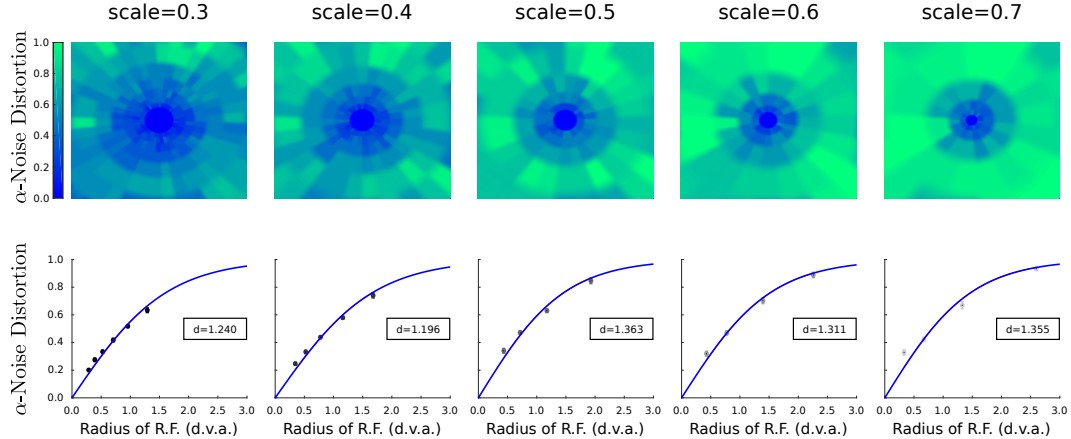

Figure 9: Top: The average $\alpha$-noise distortion over the entire visual field for our 10 images without assuming tangential homogeneity. Notice that on average, $\alpha$ increases radially. Bottom: The $\gamma(\cdot)$ which completely defines the $\alpha$-noise distortion for any receptive field as a function of its size (radius).

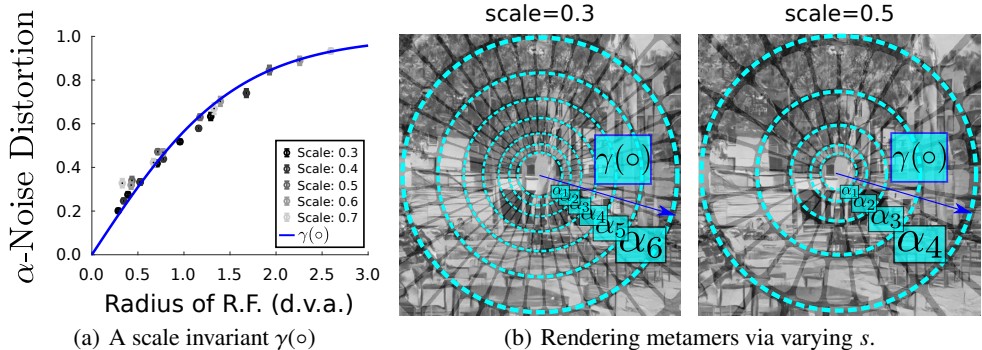

(a) A scale invariant $\gamma(\circ)$      (b) Rendering metamers via varying $s$.

Figure 10: Metamer generation proces for Experiment 2. We modulate the distortion for each receptive field according to $\gamma$ to perform an optimization as in Freeman & Simoncelli (2011).

## 4.2 EXPERIMENT 2: PSYCHOPHYSICAL EVALUATION OF METAMERISM WITH HUMAN OBSERVERS

Given that we have estimated the value of $\alpha$ anywhere in the visual field via the $\gamma$ function, we can now render our metamers as a function of the single scaling parameter ($s$), as the receptive field size $z$ is also a function of $s$ as shown in Figure 10. The psychophysical optimization procedure is now tractable on human observers and has the following form where $0 < d'(s, \gamma(z(s); s)|\theta_{obs}) < \epsilon$:

$$s_0 = \arg\max_s \mathbb{E}[d'(s, \gamma(z(s))|\theta_{obs})] \tag{10}$$

Inspired by the evaluations of Wallis et al. (2016), we wanted to test our metamers on a group of observers performing two different ABX discrimination tasks in a roving design:

1. Discriminating between Synthesized images (Synth *vs* Synth): This has been done in the original study of Freeman & Simoncelli. While this test does not gaurantee metamerism (Reference *vs* Synth), it has become a standard evaluation when probing for metamerism.

2. Discriminating between the Synthesized and Reference images (Synth *vs* Reference). This metamerism test, was not previously reported in Freeman & Simoncelli (2011) for their original images and is the most rigorous evaluation. Recently Wallis et al. (2018) argued that any model that maps an image to white noise might gaurantee metamerism under the Synth *vs* Synth condition but not against the original/reference image, thus is not a metamer.

We had a group of 3 observers agnostic to the peripheral distortions and purposes of the experiment performed an interleaved Synth vs Synth and Synth vs Reference experiment for NF metamers for the previous set of images (Fig. 6). An SR EyeLink 1000 desk mount was used to monitor their gaze for the center forced fixation ABX task as shown in Figure 11. In each trial, observers were

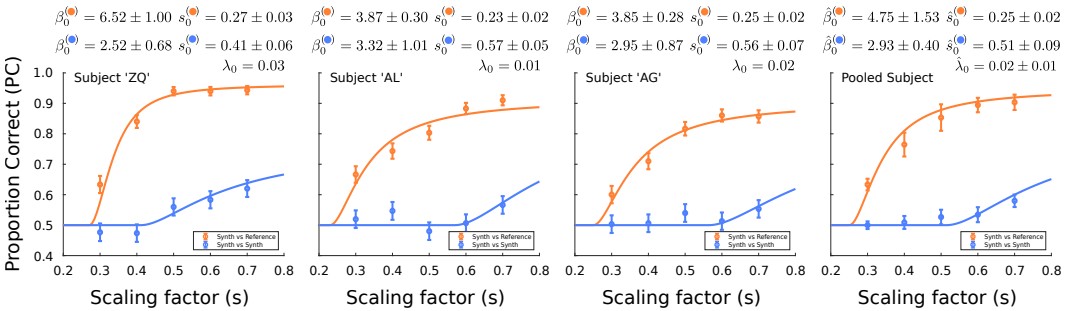

Figure 11: Experiment 2 shows the ABX metamer discrimination task done by the observers. Humans must fixate at the center of the image (no eye-movements) throughout the trial for it to be valid.

Figure 12: The results of the 3 observers and the pooled observer (average; shown on far right) for the Synth vs Reference and Synth vs Synth experiment for our metamers. The error bars denote the 68% confidence interval after bootstrapping the trials per observer.

shown 3 images where their task is to match the third image to the 1st or the 2nd. Each observer saw each of the 10 images 30 times per scaling factor (5) per discriminability type (2) totalling 3000 trials per observer. Images were rendered at $512 \times 512$ px, and we fixed the monitor at 52cm viewing distance and $800 \times 600$px resolution so that the stimuli subtended $26 \deg \times 26 \deg$. The monitor was linearly calibrated with a maximum luminance of $115.83 \pm 2.12 \ cd/m^2$. We then estimated the critical scaling factor $s_0$, and absorbing factors $\beta_0$ of the roving ABX task to fit a psychometric function for Proportion Correct (PC) as in Freeman & Simoncelli (2011); Wallis et al. (2018), where the detectability is computed via $d^2(s) = \beta_0(1 - \frac{s_0^2}{s^2})\mathbb{1}_{s > s_0}$, and

$$PC(s) = \Phi\left(\frac{d^2(s)}{\sqrt{(6)}}\right)\Phi\left(\frac{d^2(s)}{2}\right) + \Phi\left(\frac{-d^2(s)}{\sqrt{(6)}}\right)\Phi\left(\frac{-d^2(s)}{2}\right) \tag{11}$$

**Results:** Absorbing gain factors $\beta_0$ and critical scales $s_0$ per observer are shown in Figure 12, where the fits were made using a least squares curve fitting model and bootstrap sampling $n = 10000$ times to produce the 68% confidence intervals. Lapse rates ($\lambda$) were also included for robustness of fit as in Wichmann & Hill (2001). Analogous to Freeman & Simoncelli (2011), we find that the critical scaling factor is 0.51 when doing the Synth vs Synth experiment which match V2, a critical region in the brain that has been identified to respond to texture as in Long et al. (2018); Ziemba et al. (2016). This suggests that the parameters we use to capture and transfer texture statistics which are different from the correlations of a steerable pyramid decomposition as proposed in Portilla & Simoncelli (2000), might the match perceptual discrimination rates of the FS metamers. This does not imply that the models are perceptually equivalent, but it aligns with the results of Ustyuzhaninov et al. (2017) which shows that even a basis of random filters can also capture texture statistics, thus different flavors of metamer models can be created with different statistics. In addition, we find that the critical scaling factor for the Synth vs Reference experiment is less than 0.5 ($\sim 0.25$, matching V1) for the pooled observer as validated recently by Wallis et al. (2018) for their CNN synthesis and FS model for the Synth vs Reference condition.

## 5 DISCUSSION

There has been a recent surge in interest with regards to developing and testing new metamer models: The SideEye model developed by Fridman et al. (2017), uses a fully convolutional network (FCN) as in Long et al. (2015) and learns to map an input image into a Texture Tiling Model (TTM) mongrel (Rosenholtz et al. (2012)). Their end-to-end model is also feedforward like ours, but no use

of noise is incorporated in the generation pipeline making their model fully deterministic. At first glance this seems to be an advantage rather a limitation, however it limits the biological plausilibility of metameric response as the same input image should be able to create more than one metamer. Another model which has recently been proposed is the CNN synthesis model developed by Wallis et al. (2018). The CNN synthesis model is gradient-descent based and is closest in flavor to the FS model, with the difference that their texture statistics are provided by a gramian matrix of filter activations of multiple layers of a VGGNet, rather than those used in Portilla & Simoncelli (2000).

The question of whether the scaling parameter is the only parameter to be optimized for metamerism still seems to be open. This has been questioned early in Rosenholtz et al. (2012), and recently proposed and studied by Wallis et al. (2018), who suggest that metamers are driven by image content, rather than bouma's law (scaling factor). Figure 9 suggests that on average, it does seem that $\alpha$ must increase in proportion to retinal eccentricity, but this is conditioned by the image content of each receptive field. We believe that the hyperparametric nature of our model sheds some light into reconciling these two theories. Recall that in Figures (4, 8), we found that certain images can be pushed stronger in the direction of it's texturized version versus others given their location in the encoded space, the local geometry of the surface, and their projection in the perceptual space. This suggests that the average maximal distortion one can do is fixed contingent on the size of the receptive field, but we are allowed to *push further* (increase $\alpha$) for some images more than others, because the direction of the distortion lies closer to the perceptual null space (making this difference perceptually un-noticeable to the human observer). This is usually the case for regions of images that are periodic like skies, or grass. Along the same lines, we elaborate in the Supplementary Material on how our model may potentially explain why creating synthesized samples are metameric to each other at the scales of (V1;V2), but only generated samples at the scale of V1 ($s = 0.25$) are metameric to the reference image.

Our model is also different to others (FS and recently Wallis et al. (2018)) given the role of noise in the computational pipeline. The previously mentioned models used noise as an initial seed for the texture matching pipeline via gradient-descent, while we use noise as a proxy for texture distortion that is directly associated with crowding in the visual field. One could argue that the same response is achieved via both approaches, but our approach seems to be more biologically plausible at the algorithmic level. In our model an image is fed through a non-linear hierarchical system (simulated through a deep-net), and is corrupted by noise that matches the texture properties of the input image (via AdaIN). This perceptual representation is perturbed along the direction of the texture-matched patch for each receptive field, and inverting such perturbed representation results in a metamer. Figure 13 illustrates such perturbations which produce metamers when projected to a 2D subspace via the locally linear embedding (LLE) algorithm (Roweis & Saul (2000)). Indeed, the 10 encoded images do not fully overlap to each other and they are quite distant as seen in the 2D projection. However, foveated representations when perturbed with texture-like noise seem to finely tile the perceptual space, and might act as a type of *biological regularizer* for human observers who are consistently making eye-movements when processing visual information. This suggests that robust representations might be achieved in the human visual system given its foveated nature as non-uniform high-resolution imagery does not map to the same point in perceptual space.

If this holds, perceptually invariant data-augmentation schemes driven by metamerism may be a useful enhancement for artificial systems that react oddly to adversarial perturbations that exploit coarse perceptual mappings (Goodfellow et al. (2015); Tabacof & Valle (2016); Berardino et al. (2017)).

Understanding the underlying representations of metamerism in the human visual system still remains a challenge. In this paper we propose a model that emulates metameric responses via a foveated feed-forward style transfer network. We find that correctly calibrating such perturbations (a consequence of internal noise that match texture representation) in the perceptual space and inverting such encoded representation results in a metamer. Though our model is hyper-parametric in nature we propose a way to reduce the parametrization via a

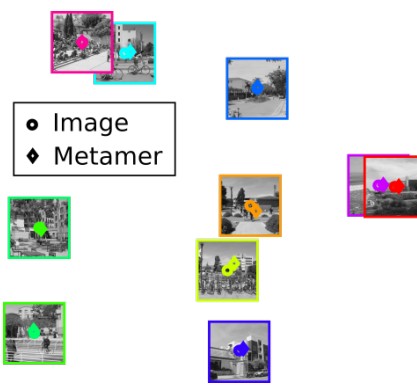

Figure 13: Image embeddings.

perceptual optimization scheme. Via a psychophysical experiment we empirically find that the critical scaling factor also matches the rate of growth of the receptive fields in V2 ($s = 0.5$) as in Freeman & Simoncelli (2011) when performing visual discrimination between synthesized metamers, and match V1 (0.25) for reference metamers similar to Wallis et al. (2018). Finally, while our choice of texture statistics and transfer is *relu*4_1 of a VGG19 and AdaIN respectively, our ×1000-fold accelerated feed-forward metamer generation pipeline should be extendible to other models that correctly compute texture/style statistics and transfer. This opens the door to rapidly generating multiple flavors of visual metamers with applications in neuroscience and computer vision.

ACKNOWLEDGEMENTS

We would like to thank Xun Huang for sharing his code and valuable suggestions on AdaIN, Jeremy Freeman for making his metamer code available, Jamie Burkes for collecting original high-quality stimuli, N.C. Puneeth for insightful conversations on texture and masking, Christian Bueno for informal lectures on homotopies, and Soorya Gopalakrishnan and Ekta Prashnani for insightful discussions. Lauren Welbourne, Mordechai Juni, Miguel Lago, and Craig Abbey were also helpful in editing the manuscript and giving positive feedback. We would also like to thank NVIDIA for donating a Titan X GPU. This work was supported by the Institute for Collaborative Biotechnologies through grant 2 W911NF-09-0001 from the U.S. Army Research Office.

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

# 6 SUPPLEMENTARY MATERIAL

Figure 14: Reference Metamers at the scale of $s = 0.25$, at which they are indiscriminable to the human observer. The color coding scheme matches the data points of the optimization in Experiment 1 and the psychophysics of Experiment 2. All images used in the experiments were generated originally at $512 \times 512$ px subtending $26 \times 26$ d.v.a (degrees of visual angle).

---

**Algorithm 1** Pipeline for Metamer hyperparameter $\gamma(\circ)$ search

---

1: **procedure** ESTIMATE HYPERPARAMETER: $\gamma(\circ)$ FUNCTION
2: Choose image dataset $S_I$.
3: Pick hyperparameter search step size $\alpha_{\text{step}}$. Pick scale search step size $s_{\text{step}}$.
4:     **for** each image $I \in S_I$ **do**
5:         **for** each scale $s \in [s_{\text{init}} : s_{\text{step}} : s_{\text{final}}]$ **do**
6:             Compute baseline metamer $M_{\text{FS}}(I)$
7:             **for** each $\alpha \in [0 : \alpha_{\text{step}} : 1]$ **do**
8:                 Compute metamer $M_{\text{NF}}(I)$
9:             **end for**
10:             Find the $\alpha$ for each receptive field that minimizes: $\mathbb{E}(\Delta\text{-SSIM})^2$.
11:             Fit the $\gamma_s(\circ)$ function to collection of $\alpha$ values.
12:         **end for**
13:     **end for**
14: Perform Permutation test on $\gamma_s$ for all $s$.
15:     **if** $\gamma_s$ is independent of $s$ **then**
16:         $\gamma_s = \gamma$
17:     **else**
18:         Perform regression of parameters of $\gamma_s$ as a function $f$ of $s$.
19:         $\gamma_s = \gamma_{f(s)}$
20:     **end if**
21: **end procedure**

---

## 6.1 HYPERPARAMETER SEARCH ALGORITHM

Algorithm 1 fully describes the outline of Experiment 1.

## 6.2 MODEL SPECIFICIATIONS AND TRAINING

We use $k = k_p + k_f$ spatial control windows, $k_p$ pooling regions ($\theta_r$ receptive fields $\times$ $\theta_t$ eccentricity rings) and $k_f = 1$ fovea (at an approximate 3 deg radius). Computing the metamers for the scales of $\{0.3, 0.4, 0.5, 0.6, 0.7\}$ required $\{300, 186, 125, 102, 90\}$ pooling regions excluding the fovea where we applied local style transfer. Details regarding the decoder network architecture and training can be seen in Huang & Belongie (2017). We used the publicly available code by Huang and Belongie for our decoder which was trained on ImageNet and a collection of publicly available paintings to learn how to invert texture as well. In their training pipeline, the encoder is fixed and the decoder is trained to learn how to invert the structure of the content image, and the texture of the style image, thus when the content and style image are the same, then the decoder approximates the inverse of the encoder ($\mathcal{D} \sim \mathcal{E}^{-1}$). We also re-trained another decoder on a set of 100 images all being scenes (as a control to check for potential differences), and achieved similar outputs (visual inspection) to the publicly available one of Huang & Belongie. The dimensionality of the input of the encoder is $1 \times 512 \times 512$, and the dimensionality of the output ($relu4\_1$) is $512 \times 64 \times 64$, it is at the $64 \times 64$ resolution that we are applying foveated pooling from the initial guidance channels of the $512 \times 512$ input.

Constructions of biologically-tuned peripheral representations are explained in detail in Freeman & Simoncelli (2011); Akbas & Eckstein (2017); Deza & Eckstein (2016), and are governed by the following equations:

$$f(x) = \begin{cases} cos^2(\frac{\pi}{2}(\frac{x-(t_0-1)/2}{t_0})); & -(1+t_0)/2 < x \le (t_0-1)/2 \\ 1; & (t_0-1)/2 < x \le (1-t_0)/2 \\ -cos^2(\frac{\pi}{2}(\frac{x-(1+t_0)/2}{t_0})) + 1; & (1-t_0)/2 < x \le (1+t_0)/2 \end{cases} \tag{12}$$

$$h_n(\theta) = f\left(\frac{\theta - (w_\theta n + \frac{w_\theta(1-t_0)}{2})}{w_\theta}\right); w_\theta = \frac{2\pi}{N_\theta}; n = 0, ..., N_\theta - 1 \tag{13}$$

$$g_n(e) = f\left(\frac{\log(e) - [\log(e_0) + w_e(n+1)]}{w_e}\right); w_e = \frac{\log(e_r) - \log(e_0)}{N_e}; n = 0, ..., N_e - 1 \tag{14}$$

where $f(x)$ is a cosine profiling function that smoothes a regular step function, and $h_n(\theta)$, $g_n(e)$, are the averaging values of the pooling region $w_i$ at a specific angle $\theta$ and radial eccentricity $e$ in the visual field. In addition we used the default values of visual radius of $e_r = 26 \deg$, and $e_0 = 0.25 \deg$ [1], and $t_0 = 1/2$. The scale $s$ defines the number of eccentricities $N_e$, as well as the number of polar pooling regions $N_\theta$ from $\langle 0, 2\pi]$. We perform the foveated pooling operation on the output of the Encoder. Since the encoder is fully convolutional with no fully connected layers, guidance channels can be used to do localized (foveated) style transfer.

Our pix2pix U-Net refinement module took 3 days to train on a Titan X GPU, and was trained with 64 crops ($256 \times 256$) per image on 100 images, including horizontally mirrored versions. We ran 200 training epochs of these 12800 images on the U-Net architecture proposed by Isola et al. (2017) which preserves local image structure given an adversarial and L2 loss.

### 6.3 METAMER MODEL COMPARISON

The following table summarizes the main similarities and differences across all current models:

| Model | FS (2011) | CNN-Synthesis (2018) | SideEye (2017) | NF (Ours) |
|---|---|---|---|---|
| Feed-Forward | - | - | ✓ | ✓ |
| Input | Noise | Noise | Image | Image |
| Multi-Resolution | ✓ | ✓ | - | - |
| Texture Statistics | Steerable Pyramid | VGG19 $conv$-$1_1, 2_1, 3_1, 4_1, 5_1$ | Steerable Pyramid | VGG19 $relu4_1$ |
| Style Transfer | Portilla & Simoncelli | Gatys et al. | Rosenholtz et al. | Huang & Belongie |
| Foveated Pooling | ✓ | ✓ | (Implicit via FCN) | ✓ |
| Decoder (trained on) | - | - | metamers/mongrels | images |
| Moveable Fovea | ✓ | ✓ | ✓ | ✓ |
| Use of Noise | Initialization | Initialization | - | Perturbation |
| Non-Deterministic | ✓ | ✓ | - | ✓ |
| Direct Computable Inverse | - | - | (Implicit via FCN) | ✓ |
| Rendering Time | hours | minutes | miliseconds | seconds |
| Image type | scenes | scenes/texture | scenes | scenes |
| Critical Scaling (*vs* Synth) | 0.46 | ~ {0.39/0.41} | Not Required | 0.5 |
| Critical Scaling (*vs* Reference) | Not Available | ~ {0.2/0.35} | Not Required | 0.24 |
| Experimental design | ABX | Oddball | - | ABX |
| Reference Image in Exp. | Metamer | Original | - | Compressed via Decoder |
| Number of Images tested | 4 | 400 | - | 10 |
| Trials per observers | ~ 1000 | ~ 1000 | - | ~ 3000 |

Table 1: Metamer Model comparison

Encoding: $\mathcal{E}(M) = \mathcal{E}(I)$

Decoding: $M = \underset{N}{\arg\min} \, ||\mathcal{E}(I) - \mathcal{E}(N)||$

Encoding: $\mathcal{E}(M) = \mathcal{E}(I) + N$

Decoding: $M = \mathcal{D}(\mathcal{E}(I) + N)$

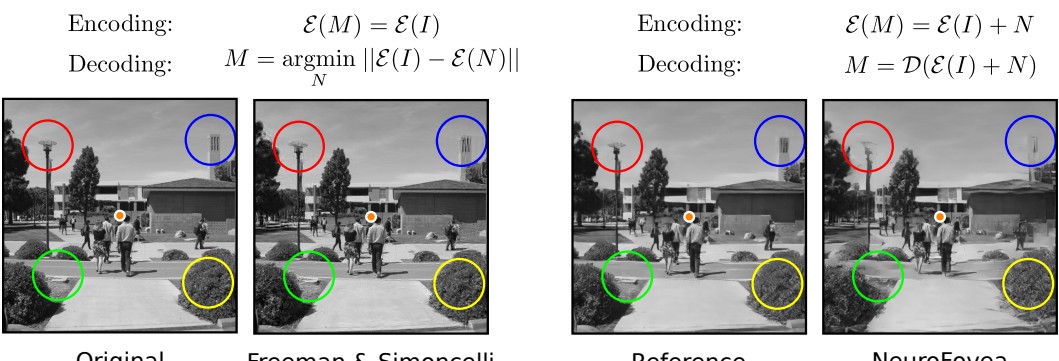

Original     Freeman & Simoncelli     Reference     NeuroFovea

Figure 15: Algorithmic (top) and visual (bottom) comparisons between our metamers and a sample from Freeman & Simoncelli (2011) for a scaling factor of 0.3. Each model has it's own limitations: The FS model can not directly compute an inverse of the encoded representation to generate a metamer, requiring an iterative gradient descent procedure. Our NF model is limited by the capacity of the encoder-decoder architecture as it does not achieve lossless compression (perfect reconstruction).

---

[1]We remove central regions with an area smaller than 100 pixels, and group them into the fovea

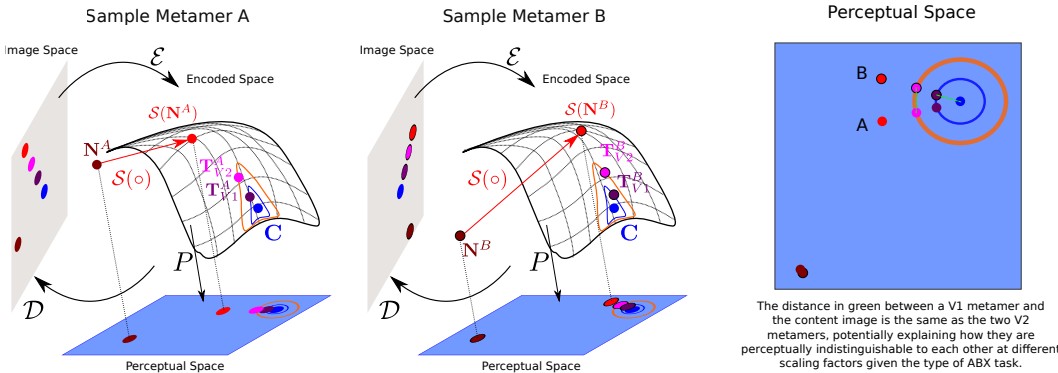

Figure 16: Decomposition and overview of the metamer generation process in the Image space, the Encoded space and the Perceptual space. The original image patch is coded in blue, the V1 metamers are coded in purple, and the V2 metamers are coded in pink. Dark brown represents the initial white noise that is later stylized via AdaIN through $\mathcal{S}(\circ)$. Note that these two points are far away to each other in image space, but quite closeby in perceptual space as they are also 'metameric' to each other. They are not placed on the actual encoded manifold since these points are not in the near vicinity of either $\mathbf{C}$ nor $\mathcal{S}(\mathbf{N})$, as they have no scene-like structure. The interpolation for maximal distortion is done along the line between $\mathbf{C}$ and $\mathcal{S}(\mathbf{N})$, these are the points in blue and red in the encoded space which represent the extremes of $\alpha = 0.0$ and $\alpha = 1.0$ respectively.

## 6.4 INTERPRETABILITY OF V1 AND V2 METAMERS

In Figure 16, we illustrate the metamer generation process for two sample metamers, given different noise perturbations. Here, we decompose Figure 4 into two separate ones for each metamer given each noise perturbation, and provide an additional visualization of the projection of the metamers in perceptual space, gaining theoretical insight on how and why metamerism arises for the synth-vs-synth condition in V2, and the synth-vs-reference condition in V1 as we demonstrated experimentally.

## 6.5 PILOT EXPERIMENTS

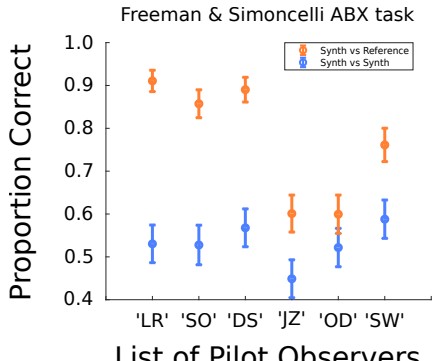

Figure 17: Pilot Data on FS metamers.

In a preliminary psychophysical study, we ran an experiment with a collection of 50 images and 6 observers on the FS metamers. Observers performed a single session of 200 trials of the FS metamers where the scale was fixed at $s = 0.5$. We found the following: While we found that the synthesized images were metameric to each other for the scaling factor of 0.5, the FS metamers were not metameric to their reference high-quality images at the scale of 0.5. Only a sub-group of observers: *'LR'*, *'SO'*, *'DS'* scored well above chance in terms of discriminating the images in the ABX task. These results are in synch with the evalutions done by Wallis et al. (2018), which varied scale and found a critical value to be less than 0.5 and rather closer to 0.25 within the range of V1.

## 6.6 Estimation of Lapse-rate ($\lambda$) per observer

The motivation behind estimating the lapse rate is to quantify how engaged was the observer in the experiment, as well as providing a robust estimate of the parameters in the fit of the psychometric functions. Not accounting for lapse rate may dramatically affect the estimation of these parameters as suggested in Wichmann & Hill (2001). In general lapse rates are computed by penalizing a psychometric function $\psi(\circ)$ that ranges between some lower bound and upper bound usually $[0,1]$. To estimate the lapse rate $\lambda$, a new $\psi'(\circ)$ is defined to have the following form:

$$\psi'(\circ) = b + (1 - b - \lambda)\psi(\circ) \tag{15}$$

Recall that for us, our psychometric fitting function $\psi(\circ) = PC_{ABX}(s)$ is defined by Equation 11 and parametrized by both the absorbing factor $\beta_0$ and the critical scaling factor $s_0$:

$$PC_{ABX}(s) = \Phi\left(\frac{d^2(s)}{\sqrt{(6)}}\right)\Phi\left(\frac{d^2(s)}{2}\right) + \Phi\left(\frac{-d^2(s)}{\sqrt{(6)}}\right)\Phi\left(\frac{-d^2(s)}{2}\right) \tag{16}$$

where we have:

$$d^2(s) = \beta_0(1 - \frac{s_o^2}{s^2})\mathbb{1}_{s > s_0} \tag{17}$$

To compute the new $\psi'(\circ)$, we notice first that our $\psi$ is bounded between $[0.5, 1]$, and that the new $\psi'$ will be a linear combination of a correct guess for a lapse, and a correct decision for a non-lapse from which we obtain:

$$PC(s) = \lambda + (1 - 2\lambda)PC_{ABX}(s) \tag{18}$$

as derived in Hénaff (2018) which includes lapse rates for an AXB task. When fitting the curves for each of the $n = 10000$ bootstrapped samples, we restricted the lapse rate to vary between $\lambda = [0.00, 0.06]$ as suggested in Wichmann & Hill (2001), and found the following lapse rates:

Observer 1: $\lambda_{ZQ}^{RS} = 0.0248 \pm 0.0209$, $\lambda_{ZQ}^{SS} = 0.0430 \pm 0.0228$.

Observer 2: $\lambda_{AL}^{RS} = 0.0008 \pm 0.0062$, $\lambda_{AL}^{SS} = 0.0166 \pm 0.0215$.

Observer 3: $\lambda_{AG}^{RS} = 0.0141 \pm 0.0243$, $\lambda_{AG}^{SS} = 0.0218 \pm 0.0236$.

We later averaged these lapse rates as there is an equal probability of each type of trial to appear (Synth vs Synth, or Reference vs Synth), and refitted each curve with the new pooled lapse rate estimates $\lambda'$. Indeed, each observer did both experiments in a roving paradigm, rather than doing one experiment after the other – thus we should only have *one* estimate for lapse rate per observer. It is worth mentioning that re-performing the fits with separate lapse rates did not significantly affect the estimates of critical scaling values, as one might argue that higher lapse rates will significantly move the critical scaling factor estimates. This is not the case as the absorbing factor $\beta$ does not place an upper bound for the psychometric function at 1.

Our critical estimates of lapse rates were: $\lambda_{ZQ} = 0.0339$, $\lambda_{AL} = 0.0087$, $\lambda_{AG} = 0.0179$, as shown in Figure 12.

The estimates (critical scale ($s_0$), absorbing factor ($\beta_0$) and lapse rate ($\lambda_0$)) shown for the pooled observer were obtained by averaging the estimates over the 3 observers.

### 6.7 ROBUSTNESS OF ESTIMATION OF $\gamma$ FUNCTION

In this subsection we show how the perceptual optimization pipeline is robust to a selection of IQA metrics such as MS-SSIM (multi-scale SSIM [2]) from Wang et al. (2003) and IW-SSIM (information content weighted SSIM) from Wang & Li (2011).

There are 3 key observations that stem from these additional results:

1. The sigmoidal natural of the $\gamma$ function is found again and is also scale independent, showing the broad applicability of our perceptual optimization scheme and how it is extendable to other IQA metrics that satisfy SSIM-like properties (upper bounded, symmetric and unique maximum).

2. The tuning curves of MS-SSIM and IW-SSIM look almost identical, given that IW-SSIM is not more than a weighted version of MS-SSIM where the weighting function is the mutual information between the encoded representations of the reference and distortion image across multiple resolutions. Differences are stronger in IW-SSIM when the region over which it is evaluated is quite large (*i.e.* an entire image), however given that our pooling regions are quite small in size, the IW-SSIM score asymptotes to the MS-SSIM score. In addition both scores converge to very similar values given that we are averaging these scores over the images and over all the pooling regions that lie within the same eccentricity ring. We found that $\sim 90\%$ of the maximum $\alpha$'s had the same values given the 20 point sampling grid that we use in our optimization. Perhaps a different selection of IW hyperparameters (we used the default set), finer sampling schemes for the optimal value search, as well as averaging over more images, may produce visible differences between both metrics.

3. The sigmoidal slope is smaller for both IW-SSIM and MS-SSIM *vs* SSIM, which yields more conservative distortions (as $\alpha$ is smaller for each receptive field). This implies that the model can still create metamers at the estimated found scaling factors of 0.21 and 0.50, however they may have different *critical* scaling factors for the reference vs synth experiment, and for the synth vs synth experiment. Future work should focus on psychophysically finding these critical scaling factors, and if they still are within the range of rate of growth of receptive field sizes of V1 and V2.

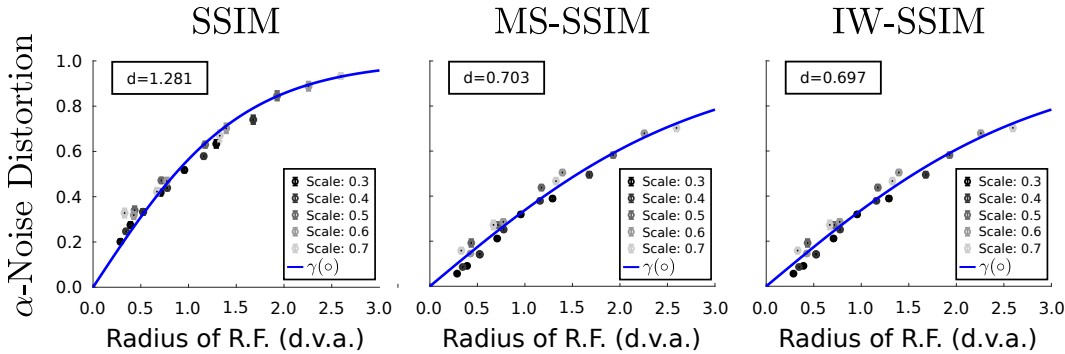

Figure 18: A collection of scale invariant $\gamma(\circ)$'s across multiple IQA metrics for the perceptual optimization scheme of Experiment 1. In this figure we superimpose all maximal $\alpha$-noise distortions for each scale, and find a function that fits all the points showing that $\gamma$ is indepedent of scale.

---

[2]scale in the context of SSIM is referred to resolution (as in scales of a laplacian pyramid), and is not to be confused with the scaling factor $s$ of our experiments which encode the rate of groth of the receptive fields.

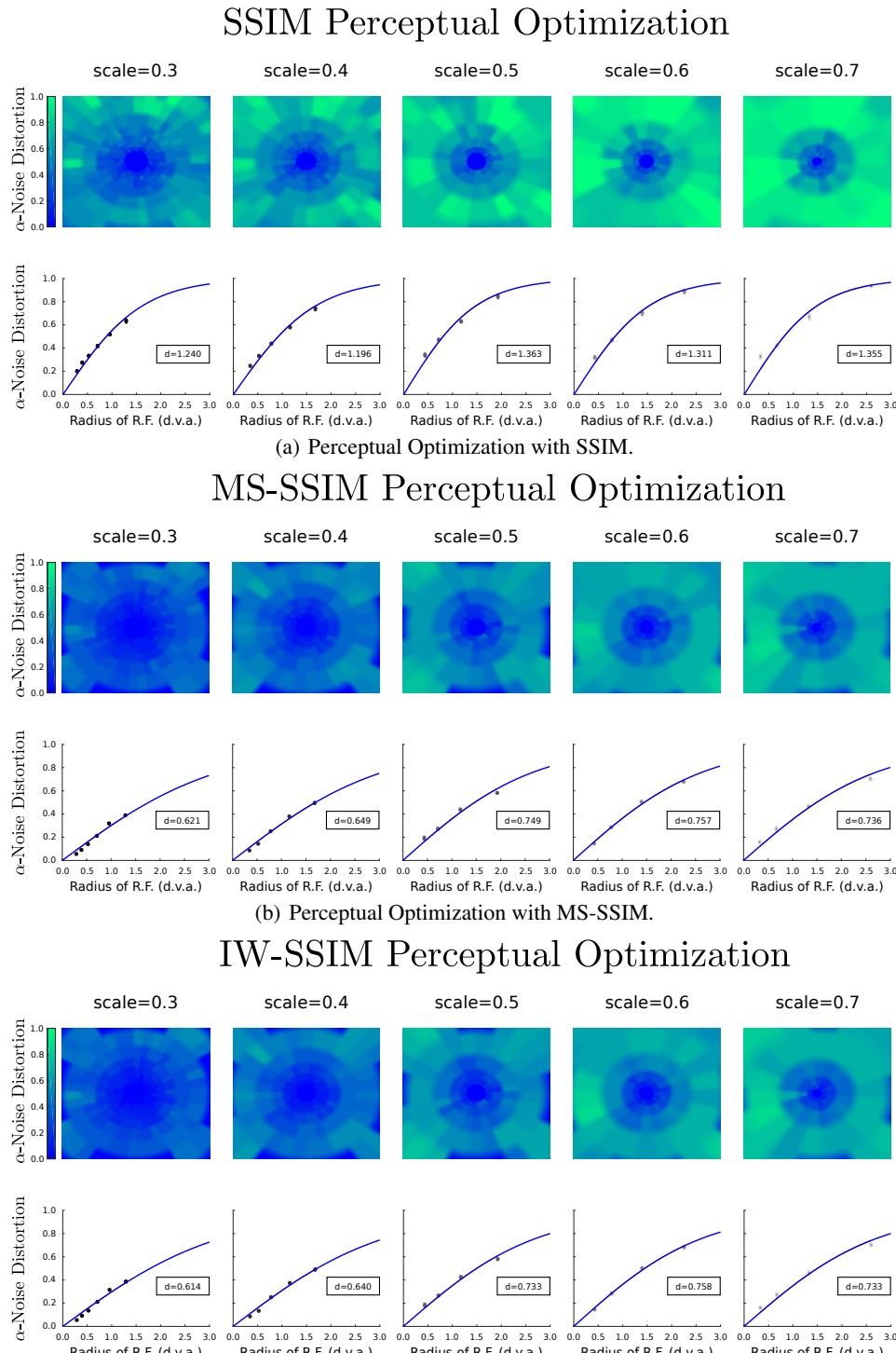

Figure 19: Top: The maximum $\alpha$-noise distortion computed per pooling region, and collapsed over all images for each IQA metric. Bottom: When averaging across all the pooling regions for each retinal eccentricity, we find that the $\gamma$ function is invariant to scale as in our original experiment – suggesting that our perceptual optimization scheme is flexible across IQA metrics.

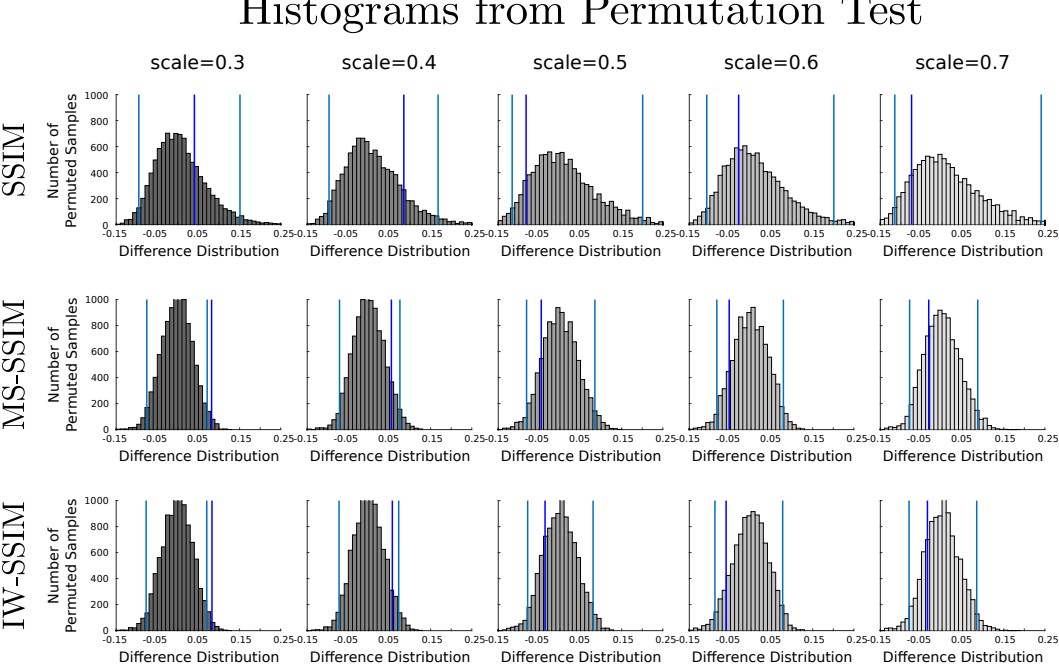

Figure 20: A permutation test was ran and determined that each $\gamma$ function is also scale independent under the 99% confidence interval (CI), as we increased the CI to account for false discovery rates (FDR). Indeed, when we perform the permutation tests and use a 95% confidence interval (shown in the figure with the vertical lines in cyan), all curves except for MS-SSIM and IW-SSIM only for the scaling factor of 0.3 show a significant difference $p \sim 0.02$ (non FDR-corrected), potentially due to small receptive field sizes, which bias the estimates. All other differences in the $d$ parameter of the sigmoid function, with respect to the average fitted sigmoid, are statistically insignificant.

