# OpenReview forum: "Towards Metamerism via Foveated Style Transfer"
_ICLR.cc/2019/Conference_

### Official Review · AnonReviewer1 · 2018-11-02
**Somewhat obscure writing, but reasonable contribution**

**Rating:** 7
**Confidence:** 5

**Review:**

Summary:
The paper proposes a fast method for generating visual metamers – physically different images that cannot be told apart from an original – via foveated, fast, arbitrary style transfer. The method achieves the same goal as an earlier approach (Freeman & Simoncelli 2011): locally texturizing images in pooling regions that increase with eccentricity, but is orders of magnitude faster. The authors perform a psychophysical evaluation to test how (in)discriminable their synthesized images are amongst each other and compared with originals. Their experiment replicates the result of Freeman & Simoncelli of a V2-like critical scaling in the synth vs. synth condition, but shows that V1-like or smaller scaling is necessary for the original vs. synth condition.

I reviewed an earlier version of this paper for a different venue, where I recommended rejection. The authors have since addressed some of my concerns, which is why I am more positive about the paper now.

Strengths:
+ The motivation for the work is clear and the implementation straightforward, combining existing tools from style transfer in a novel way.
+ It's fast. Rendering speed is indeed a bottleneck in existing methods, so a fast method is useful.
+ The perceptual quality of the rendered images is quantified by psychophysical testing.
+ The role of the scaling factor for the pooling regions is investigated and the key result of Freeman & Simoncelli (pooling regions scale with 0.5*eccentricity) is replicated with the new method. In addition, the result of Wallis et al. (2018) that lower scale factors are required for original vs. synth is replicated as well.


Weaknesses:
- Compared with earlier work, an additional fudge parameter (alpha) is introduced. It is not clear why it is necessary and it complicates interpretation.
- The paper contains a number of sections with obscure mathiness and figures that I can't follow and whose significance is unclear.


Conclusion:
The work is well motivated, the method holds up to its promise of being fast and is empirically validated. However, it feels quite ad-hoc and the writing of the paper is very obscure at various places, which leaves room for improvement.


Details:

- The motivation for introducing alpha not clear to me. Wasn't the idea of F&S that you can reduce the image to its summary statistics within a pooling region whose size scales with eccentricity? Why do you need to retain some content information in the first place? How do images with alpha=1 (i.e. keep only texture) look?

- Related to above, why does alpha need to change with eccentricity? Experiment 1 seems to suggest that changing alpha leads to similar SSIM differences between synths and originals as F&S does, but what's the evidence that SSIM is a useful/important metric here?

- Again related to above, why do you not use the same approach of blending pooling regions like F&S did instead of introducing alpha?

- I would like to know some details about the inference of the critical scaling. It seems surprisingly spot on 0.5 as in F&S for synth vs. synth, but looking at the data in Fig. 12 (rightmost panel), I find the value 0.5 highly surprising given that all the blue points lie more or less on a straight line and the point at a scaling factor of 0.5 is clearly above chance level. Similarly, the fit for original vs. synth does not seem to fit the data all that well and a substantially shallower slope seems equally plausible given the data. How reliable are these estimates, what are the confidence intervals, and was a lapse rate included in the fits (see Wichmann & Hill 2001)?

- I don't get the point of Figs. 4, 13 and 14. I think they could as well be removed without the paper losing anything. Similarly, I don't think sections 2.1 and the lengthy discussion (section 5) are useful at all. Moreover, section 3 seems bogus. I don't understand the arguments made here, especially because the obvious options (alpha=1 or overlapping pooling regions; see above) are not even mentioned.

- How is the model trained? Do the authors use the pre-trained model of Huang & Belongie or is the training different in the context of the proposed method? I could only find the statement that the decoder is trained to invert the encoder, but that doesn't seem to be what Huang & Belongie's model does and the paper does not say anything about how it's trained to invert. Please clarify.

- At various places the writing is somewhat sloppy (missing words, commas, broken sentences), which could have been avoided by carefully proof-reading the paper.

---

> ### Author Response · Authors · 2018-11-21
> **Comments to AnonReviewer 1 [Part 1]**
>
> Thank you for providing critical insights in our paper and giving such positive feedback! We would like to address your concerns:
>
> - The motivation for introducing alpha not clear to me. Wasn't the idea of F&S that you can reduce the image to its summary statistics within a pooling region whose size scales with eccentricity? Why do you need to retain some content information in the first place? How do images with alpha=1 (i.e. keep only texture) look?
>
> You are correct on the idea that F&S introduce the texture matching hypothesis for peripheral processing (as preceded by Balas et al., 2009), however while the original work of Balas et al. 2009 uses pure texture matching with no structural constraints to explain losses in performance for a visual search task, the model and implementation of FS includes a prior of image structure at each step when performing gradient descent for texture matching. You could think of this as globally trying to minimize the mean square error (MSE) between the initial noise seed and the final image in pixel space, and locally at the same time, they are matching the MSE in texture space (via the Portilla-Simoncelli statistics) between the initial noise seed and the image content that lies within each receptive field. Images with pure alpha=1 for each receptive field show highly aberrant distortions, as would FS metamers if the content/structure matching restriction would not be added in their implementation. Consequently, a pure texture matching approach does not work for visual metamerism. This has been clarified with great detail in Wallis et al. (Journal of Vision) 2016 -- See Figure 7., General Discussion (Texture Statistics & Metamerism), and Acknowledgements of their paper, and also been suggested recently in Wallis, Funke et al., 2018. In addition the pioneering work of Rosenholtz et al. 2012 (Journal of Vision) on Mongrels as well as the Texforms of Long, Yu & Konkle (PNAS, 2018) provide the same intuitions and clarifications with regards to preserving structure. It is a subtle detail present in the original FS code, and might have not been emphasized in the original paper.
>
> Here is a link to the line in the code, where they project the image to its low pass residual (a way of enforcing structural constraints) for every step of the texture matching procedure that is done via a set of coarse-to-fine sub-iterations:
>  https://github.com/freeman-lab/metamers/blob/master/main/metamerSynthesis.m#L191
>
> We would really like to thank you for pointing this out, as it is a detail that if not properly addressed, defeats the whole purpose of trying to preserve image structure -- and introducing an alpha parameter in the first place. Hopefully we have addressed your main concern, and appreciate the rigorous feedback that has propelled this work forward from previous versions.
>
> - Related to above, why does alpha need to change with eccentricity? Experiment 1 seems to suggest that changing alpha leads to similar SSIM differences between synths and originals as F&S does, but what's the evidence that SSIM is a useful/important metric here?
>
> Alpha should change as a function of eccentricity given higher effects of crowding. We empirically verified this is the case by fitting a gamma function that tunes each alpha coefficient as a function of receptive field size which increase with eccentricity. With regards to the choice of SSIM over other IQA metrics, please see our detailed response to AnonReviewer 3 who has suggested trying Experiment 1 with other IQA metrics. We have done so, finding that the tuning properties of the gamma function still hold, and have added these results in the updated Supplementary Material (Section 6.7) .
>
> - Again related to above, why do you not use the same approach of blending pooling regions like F&S did instead of introducing alpha?
>
> We do indeed use blended pooling regions as in F&S, and would like to clarify that the interpolation in Figures 3 and 4 are done for each pooling region, rather than the whole image. You could think of Figure 3 as a ‘zoomed in’ pooling region as we wanted to magnify the effects of the distortions within a receptive field. These smoothly blended pooling regions are used for local style transfer for each receptive field. Figure 9 (top), shows how we assign an alpha coefficient to each pooling regions (receptive field) and Section 6.2 in the supplementary material provides details on the construction of blended pooling regions.

---

> > ### Author Response · Authors · 2018-11-21
> > **Comments to AnonReviewer 1 [Part 2]**
> >
> > - I would like to know some details about the inference of the critical scaling. It seems surprisingly spot on 0.5 as in F&S for synth vs. synth, but looking at the data in Fig. 12 (rightmost panel), I find the value 0.5 highly surprising given that all the blue points lie more or less on a straight line and the point at a scaling factor of 0.5 is clearly above chance level. Similarly, the fit for original vs. synth does not seem to fit the data all that well and a substantially shallower slope seems equally plausible given the data. How reliable are these estimates, what are the confidence intervals, and was a lapse rate included in the fits (see Wichmann & Hill 2001)?
> >
> > We followed your suggestion and reported the lapse rates in the updated manuscript (all under 3%). There was little variability in the fits, as the absorbing factor generally takes care of modulating the asymptotic performance of the psychometric function for each subtack in our roving experiment. We elaborate more on the fitting procedure as well as the characteristics of the psychometric function and lapse rates in the Supplementary Material. We also included confidence intervals for all the estimates of critical scaling factor, absorbing factor and lapse rates in our updated version (See Figure 12).
> >
> > You are correct. In order to compute the average fitted values for the pooled observer, we averaged the fitted values for the 3 observers and labelled them as the average fit, rather than performing least squares regression on the average values (which was done individually).
> >
> > We added these clarifications in the supplementary material of paper, as well as a derivation on how to compute the lapse rate for our ABX task.
> >
> > - I don't get the point of Figs. 4, 13 and 14. I think they could as well be removed without the paper losing anything. Similarly, I don't think sections 2.1 and the lengthy discussion (section 5) are useful at all. Moreover, section 3 seems bogus. I don't understand the arguments made here, especially because the obvious options (alpha=1 or overlapping pooling regions; see above) are not even mentioned.
> >
> > We moved Figure 13 to the Supplementary Material.
> >
> > We believe Figures 4, and 14 provide a clear interpretation of the model in terms of distortions in the encoded space and how distortions when viewed by the human observer change as a function of alpha and the geometry of the surface in the encoded space. We think there is a lot of room and work to do in terms of integrating the literature of visual metamerism within the context of differential geometry -- which is outside of the scope of this paper, but we provide hints on why it could be appropriate, and is currently being developed as a follow up work. The recent work of Henaff (2018) on the perceptual straightening hypothesis, as well as Eidolon distortions by Koenderink et al. (2017), are both examples of integrations between vision science and differential geometry.
> >
> > Section 3 provides mathematical insight of the psychophysical tractability of the metamer rendering problem, given that a structural constraint should be included as we clarified previously. For example, a per-pooling region tuning of maximal distortion is psychophysically intractable given that the experimenter would have to explore all pooling regions, over many values of alpha and over a wide collection of images, over many scales and for multiple trials. If we assume (similar to our settings): 100 pooling regions, 5 scales, 10 steps for alpha, 10 images, 30 trials, and 2 seconds per trial for observer response this amounts to roughly 1 month of raw psychophysics time. If we take into account that observers usually do a maximum of 2 hours per day -- this would extrapolate to a year in actual data collection time. It is possible, but unreasonable. This is the benefit of the perceptual optimization simulated experiment we propose.
> >
> > Section 3 also provides formality of the psychophysical optimization to be performed to find the critical scaling value for the FS metamers and motivates the need for Experiment 1, where we reduce the hyper-parametric nature of our model to a single parameter.

---

> > > ### Author Response · Authors · 2018-11-21
> > > **Comments to AnonReviewer 1 [Part 3]**
> > >
> > > - How is the model trained? Do the authors use the pre-trained model of Huang & Belongie or is the training different in the context of the proposed method? I could only find the statement that the decoder is trained to invert the encoder, but that doesn't seem to be what Huang & Belongie's model does and the paper does not say anything about how it's trained to invert. Please clarify.
> > >
> > > We use the pre-trained decoder by Huang and Belongie as stated in our paper. The decoder does invert the encoder with high fidelity if the input images for style and content are the same, both in theory (all the statistics are matched in the VGG19), and in practice (visual inspection and alpha=0 SSIM scores as reported in the paper). In the training pipeline, the encoder is fixed and the decoder is trained to learn how to invert the structure of the content image, and the texture of the style image, thus when the content and style image are the same, then the decoder approximates the inverse of the encoder. In the supplementary material we provide details of such training as uploaded in our submission, the content images were natural scenes from ImageNet and the style images were a collection of paintings and texture-like images.
> > >
> > > We would like to emphasize as stated in our paper that within our pipeline, there is no explicit training to render metamers, but rather to invert image structure and texture-driven distortions in the encoded space.

---

### Official Review · AnonReviewer3 · 2018-11-05
**Review of Towards Metamerism via Foveated Style Transfer**

**Rating:** 8
**Confidence:** 4

**Review:**

This paper presents an interesting analysis of metamerism and a model capable of rapidly producing metamers of value for experimental psychophysics and other domains.

Overall I found this work to be well written and executed and the experiments thorough. Specific points on positives and negatives of the work follow:

Positives:
- The paper shows a solid understanding of the literature in this domain and presents a strong motivation
- The problem itself is addressed at a deep level with many nuanced (but important) considerations discussed
- Ultimately the results of the model seem convincing in particular with the accompanying psychophysical experiments

Negatives:
- (Maybe not a negative, but a question) At the extreme tradeoff between intrinsic structure and texture, the notion of a metamer seems somewhat obscured. At what point is a metamer no longer a metamer?
- (Also not necessarily a negative) Exercising SSIM is a valid decision given it's widespread use. I am curious if MS-SSIM, IW-SSIM or other metrics make any significant difference.

---

> ### Author Response · Authors · 2018-11-21
> **Comments to AnonReviewer 3 [Part 1]**
>
> Thank you for having a very positive outlook on our paper, we will address some of your comments and questions
>
> --- At the extreme tradeoff between intrinsic structure and texture, the notion of a metamer seems somewhat obscured. At what point is a metamer no longer a metamer?
>
> This is a great question. In general, two stimuli are metameric to each other when they are perceptually indistinguishable, under certain viewing conditions. In our experiments the viewing condition is restricted to a forced fixation task at the center of each image. To answer your question, this happens when the scaling value that is used to construct the size of the pooling regions exceeds their critical limit. All images below such critical scaling values remain metameric to each other contingent on the testing paradigm: Reference vs Synthesis (s=0.25) and Synthesis vs Synthesis Experiment (s=0.5). Indeed, you could imagine a small alteration in an image, such as modifying a specific pixel by 1 bit, that could also produce a metamer. Yet that distortion is somewhat uninteresting, and most importantly it does not provide theoretical insights on the computations done by the human visual system (texture matching in the periphery as proposed in Balas et al, 2009 and Freeman and Simoncelli 2011). Moreover, we find a function (the gamma function), that modulates how much distortion (quantified by alpha) to insert contingent on the size of each receptive field, for any scaling factor. Figure 4 illustrates this idea with the blue contour around the blue dot which we call the metameric boundary, if a distortion exceeds such value, the synthesized image will fail to be metameric locally for a receptive field, and thus for the entire image.

---

> > ### Author Response · Authors · 2018-11-21
> > **Comments to AnonReviewer 3 [Part 2]**
> >
> > --- (Also not necessarily a negative) Exercising SSIM is a valid decision given it's widespread use. I am curious if MS-SSIM, IW-SSIM or other metrics make any significant difference.
> >
> > This is also a great observation. In principle we chose SSIM because it is has been empirically been shown to be monotonic with human judgments of visual perception in terms of distortions. Other important factors of our choice of SSIM that we did not include in the paper, is that SSIM is based on changes of luminance, contrast, and structure (via normalized contrast), all of these which are critical aspects when analyzing distortions. In addition, SSIM is upper bounded, symmetric and has a unique maximum; which are all ideal traits to have for the perceptual optimization pipeline proposed in Experiment 1 (Section 4.1). MS-SSIM (multiscale SSIM) and IW-SSIM (image content weighted SSIM, computed via mutual information between the encoded reference and distorted image) also share these properties and following your suggestion we decided to re-run Experiment 1 with these IQA metrics to analyze the robustness of choice for SSIM vs other metrics as well as to analyze the potential change of shape of the gamma function. This experiment served as a great control: as it showed that our optimization scheme is extendible to other IQA metrics. (See Algorithm 1 in the Supplementary Material in our original and updated submission).
> >
> > We have added a page in the Supplementary Material (Section 6.7), with such updates results, figures and permutation tests, and where we discuss our updated results and what we found. We have copied them here:
> >
> > There are the 3 key observations that stem from these additional results:
> >
> > 1) The sigmoidal natural of the gamma function is found again and is also scale independent, showing the broad applicability of our perceptual optimization scheme and how it is extendable to other IQA metrics that satisfy SSIM-like properties (upper bounded, symmetric and unique maximum).
> >
> > 2) The tuning curves of MS-SSIM and IW-SSIM look almost identical, given that IW-SSIM is not more than a weighted version of MS-SSIM where the weighting function is the mutual information between the encoded representations of the reference and distortion image across multiple resolutions. Differences are stronger in IW-SSIM when the region over which it is evaluated is quite large (i.e. an entire image), however given that our pooling regions are quite small in size, the IW-SSIM score asymptotes to the MS-SSIM score. In addition both scores converge to very similar values given that we are averaging these scores over the images and over all the pooling regions that lie within the same eccentricity ring. We found that ~90% of the maximum alpha's had the same values given the 20 point sampling grid that we use in our optimization. Perhaps a different selection of IW hyperparameters (we used the default set), finer sampling schemes for the optimal value search, as well as averaging over more images, may produce visible differences between both metrics.
> >
> > 3) The sigmoidal slope is smaller for both IW-SSIM and MS-SSIM vs SSIM, which yields more conservative distortions (as alpha is smaller for each receptive field). This implies that the model can still create metamers but potentially with different critical scaling factors for the reference vs synth experiment, and for the synth vs synth experiment. Future work should focus on psychophysically finding these critical scaling factors, and if they still are within the range of rate of growth of receptive field sizes of V1 and V2.

---

### Official Review · AnonReviewer4 · 2018-11-10
**Towards Metamerism via Foveated Style Transfer**

**Rating:** 7
**Confidence:** 4

**Review:**

Summary
This paper proposes a NeuroFovea (NF) model for generation of point-of-fixation metamers. As opposed to previous algorithms which use gradient descent to match the local texture and image statistics, NF proposed to use a style transfer approach via an Encoder-Decoder style architecture, which allows it to produce metamers in a single forward pass, allowing it to achieve a significant speed-up as compared to early approaches.

Pros
-The paper tackles a very intriguing topic.
-The paper is very well written using concise and clear language allowing it to present a large -amount of information in the 10 pages + appendix.
-The paper provides a thorough discussion of both the problem, related work and the model itself.
-A single forward pass nature of the model allows it to achieve a 1000x speed-up in generating metamers as opposed to previous GD based approaches.
-The authors provide enough details to allow for reproducibility.

Cons
-(Not necessarily a negative) Requires a very careful reading as the paper provides a lot of information (though as mentioned it is very well written)
-The quantitative evaluation is somewhat lacking in that there are no quantitative psychophysical experiments to compare this model to competing ones across different observers. For example, it would have been interesting to compare the ability of observers to distinguish between original images and metamers generated by different models.

Additional comments
On page 10., you show Fig. 13 however you mention at the end of the first paragraph you further elaborate on Fig 13. in the Supplementary Materials. I think it would be better to either provide more discussion in the text and refer to the figure, or just move it fully to Supplementary materials.

Also, in the qualitative comparison of various models you mention that SideEye runs in milliseconds whereas NF runs in seconds. It would be interesting to discuss the potential trade-off between speed and the quality of generated metamers between the models.

---

> ### Author Response · Authors · 2018-11-21
> **Comments to AnonReviewer 4**
>
> Thanks for taking the time to review our paper. We also share your enthusiasm with regards to metamerism. Below we address some of the comments:
>
> --- The quantitative evaluation is somewhat lacking in that there are no quantitative psychophysical experiments to compare this model to competing ones across different observers. For example, it would have been interesting to compare the ability of observers to distinguish between original images and metamers generated by different models.
>
> This is an excellent point and we are currently working towards that direction. The current submission represents a good first step: to fully describe our model, and psychophysically evaluate it under 2 conditions (synth vs synth, and synth vs reference). A next step is to evaluate our model with other models including the FS for the same set of images. One current limitation when considering such rigorous evaluation, is that both the SideEye model and the CNN Synthesis model are not publicly available -- thus the differences in performance might be driven by hyperparameter/implementation settings for each model, rather by the model itself. Along these lines, we are looking forward to release our code and make it public similar to the FS model, to promote the development of improved metamer generation models as well as to see potential applications of metamerism in computer vision as suggested in the discussion section.
>
> --- Additional comments: On page 10., you show Fig. 13 however you mention at the end of the first paragraph you further elaborate on Fig 13. in the Supplementary Materials. I think it would be better to either provide more discussion in the text and refer to the figure, or just move it fully to Supplementary materials.
>
> Thanks for pointing this out. We moved Figure 13 to the Supplementary Material where we elaborate more on the geometrical interpretation of these distortions in the encoded space and how a human observer might not be able to discriminate between such distortions.
>
> --- Additional comments: Also, in the qualitative comparison of various models you mention that SideEye runs in milliseconds whereas NF runs in seconds. It would be interesting to discuss the potential trade-off between speed and the quality of generated metamers between the models.
>
> We agree, and this goes back to the point we mentioned earlier with regards to publicly available code from the authors. One of the main differences that we can comment on, is that they differ in distortions given the difference in texture statistics. We have verified this via visual inspection. The SideEye model uses a Fully Convolutional Network to approximate a Texture Tiling Model (Mongrel) in O(1) time that locally matches texture distortions everywhere in the field analogous to the metamers of FS.  These Mongrels use Portilla Simoncelli texture statistics as compared to the output of the VGG-Net that we use in our parametrization.
>
> Comparing all models is a next step in metamer research, and we will begin conversations with some of the other authors to see if we can share/distribute our code for such comparisons. In addition, the work of Wallis, Funke et al., 2018 has also shown that the choice of evaluations on images (texture-like, scene-like and man-made) also affects the difficulty of metameric rendering.  Thus, the field is not  only limited by access to models, and the code, but also by the lack of a standardized set of images and psychophysical paradigm for evaluation.

---

### Author Response · Authors · 2018-11-21
**General Comments to All Reviewers**

We’d like thank all reviewers for the feedback and assessment of our paper. We hope to have individually addressed all your concerns. We have uploaded a modified version of our paper where we have addresses such concerns, re-arranged figures, and fixed minor typos and corrections. These include:

Moving Figure 13 to the Supplementary Material for a detailed discussion on the potential interpretability of V1 and V2 metamers in the human visual system.

Enhancing Figure 4 with the subfigures of Figure 3 where we show how the interpolation is done in the encoded space.

Adding the clarification in Section 3 that the FS model includes structural constraints in the metamer generation pipeline as shown in Wallis et al. 2016.

An extended version of Figure 9 where we tune a gamma function via the perceptual optimization framework of Experiment 1, but using other IQA metrics such as MS-SSIM and IW-SSIM. This figure is supplementary and added in the Supplementary Material.

The histograms of the permutation tests that verify the scale invariance of the gamma function.

Confidence intervals for the estimates of critical scaling and absorbing factors, as well as the lapse rate for each observer in Experiment 2.

---

### Meta-Review · Area_Chair1 · 2018-12-13
**novel high-performing model; thorough experimental analysis and discussion; clarity could be improved**

**Confidence:** 4
**Recommendation:** Accept (Poster)

**Metareview:**

1. Describe the strengths of the paper.  As pointed out by the reviewers and based on your expert opinion.

- The problem is well-motivated and related work is thoroughly discussed
- The evaluation is compelling and extensive.

2. Describe the weaknesses of the paper. As pointed out by the reviewers and based on your expert opinion. Be sure to indicate which weaknesses are seen as salient for the decision (i.e., potential critical flaws), as opposed to weaknesses that the authors can likely fix in a revision.

- Very dense. Clarity could be improved in some sections.

3. Discuss any major points of contention. As raised by the authors or reviewers in the discussion, and how these might have influenced the decision. If the authors provide a rebuttal to a potential reviewer concern, it’s a good idea to acknowledge this and note whether it influenced the final decision or not. This makes sure that author responses are addressed adequately.

No major points of contention.

4. If consensus was reached, say so. Otherwise, explain what the source of reviewer disagreement was and why the decision on the paper aligns with one set of reviewers or another.

The reviewers reached a consensus that the paper should be accepted.